# A Single-Incision Method for the Removal of Vagus Nerve Stimulators: A Single-Institution Retrospective Review

**DOI:** 10.3390/brainsci15070738

**Published:** 2025-07-10

**Authors:** Michael Baumgartner, Matthew Diehl, James E. Baumgartner

**Affiliations:** 1Department of Neurological Surgery, University of California San Francisco, San Francisco, CA 94143, USA; michael.baumgartner@ucsf.edu; 2Department of Children’s Surgery, AdventHealth Orlando, Orlando, FL 32803, USA; mpdiehl99@yahoo.com; 3Department of Neurosurgery, College of Medicine, University of Central Florida, Orlando, FL 32827, USA

**Keywords:** vagal nerve stimulation, VNS removal, single incision, transient vocal cord weakness, intractable epilepsy

## Abstract

Vagal nerve stimulators (VNSs) improve seizure control in up to half of the patients who have them implanted. In non-responding patients, VNS removal may be necessary. Removal is traditionally accomplished through two incisions. We present our experience removing VNSs through a single incision. **Background/Objectives**: To determine if VNS removal can be safely performed through a single incision. **Methods**: The medical records of 73 consecutive patients who underwent VNS removal at our institution from 2012 to 2024 were reviewed. Patients were divided into single-incision and two-incision treatment groups. Operative time and surgical complications were compared between groups. **Results**: A total of 73 patients underwent VNS removal during the study timeframe. Forty-eight VNS removals were accomplished via a single incision, while 25 required both incisions. Time in the operating room was roughly half as long for single-incision removal vs. two-incision removal (29.4 min, range 11–84 vs. 74.2 min, range 33–203); however, single incision was initially attempted in all cases. In two of the incision cases, the neck dissection resulted in an injury to the internal jugular (IJ) vein. In one case, the IJ was repaired and the lead wire removed. In a second case, the IJ could not be repaired, and a segment of lead wire was retained. In a third case, a short length of lead wire was discovered after a single-incision removal and a second procedure was necessary for removal. There were no significant differences in the rates of transient vocal cord weakness, cough, and/or dysphagia between both treatment groups (*p* = 0.7368), and there were no cases of permanent nerve palsy. **Conclusions**: VNS removal can be safely accomplished via a single incision in most cases. Successful single-incision procedures may be shorter than the two-incision approach. Attempted VNS removal via a single incision may result in increased incidence of transient hoarseness, dysphagia, and/or cough, but may result in reduced rates of permanent injury or IJ injury.

## 1. Introduction

Vagal nerve stimulation is the first neurostimulation therapy approved for the treatment of drug-resistant epilepsy. Blinded randomized controlled studies have shown that 23–57% of patients with intractable epilepsy achieve at least a 50% seizure reduction with VNS therapy in the short term [1,2,3,4,5]. Longer-term seizure outcome studies suggest that response rates to vagal nerve stimulation improve with chronic stimulation in responding subjects [1,4,6,7].

Although the response rate to VNS therapy is relatively high, many patients achieve little or no improvement in their epilepsy. Many of these patients request VNS removal. Furthermore, VNSs may be removed because it interferes with subsequent epilepsy evaluations, including magnetoencephalography and MRI, as well as treatments that rely on intra-procedure MRI. While the removal of the VNS generator is straightforward, the removal of the VNS lead from the neck carries risk of injuring the recurrent laryngeal nerve and the vessels of the carotid sheath. This dissection is considerably more challenging than during implantation due to scarring at the site and the loss of anatomical landmarks and planes. A meta-analysis of pediatric VNS explantation revealed that when a two-incision VNS removal is performed, vascular injury occurs in 1% of cases, transient vocal cord paresis/paralysis, dysphonia, and dysphagia occur in 3.6% of cases, and permanent injury occurs in 1.5% of cases. These complications are attributed to the neck dissection [8].

The VNS device consists of a generator implanted through a chest wall incision and a lead wire inserted via a neck incision. The lead wire has helical stimulating coils that are placed circumferentially around the vagal nerve, and the lead wire is then tunneled to reach the generator [1]. As per the Liva Nova Physicians Manual (2022), MRI should not be performed if more than 2 cm of lead wire remains in the patient’s chest wall or neck due to the risk of it heating and causing thermal injury. The helical coils, however, are not a contraindication against MRI. During standard two-incision VNS removal procedures, we noted that the VNS wire sometimes fractured with less than 2 cm of lead wire remaining after the application of moderate traction without resulting vascular injury or postoperative dysphagia, dysphonia, or hoarseness. We sought to explore if this finding could be exploited in the operating room to avoid neck incision, and we therefore tested whether the entire lead wire could be removed by fracturing the lead wire/helical coil connection via the application of traction on the exposed electrode at the generator/chest wall incision, along with countertraction at the site of lead placement in the neck.

## 2. Materials and Methods

### 2.1. Study Design

After obtaining permission from the Advent Health IRB, the charts of all patients who underwent attempts to remove the VNS lead through a single incision from 2012 to 2024 were retrospectively reviewed. All patients were evaluated by our epilepsy neurology team and the decision for VNS explant was reviewed and approved through our Comprehensive Epilepsy Case Conference. The chart review included but was not limited to surgical technique, duration of surgery, the presence or absence of vocal cord weakness, need for transfusion, injury to the internal jugular vein, and duration of vocal cord weakness postoperatively. The duration of surgery data was only available for patients who underwent VNS removal. This is because when the VNS removal was performed and then followed by a second procedure during the same OR visit, no distinction was made between the OR time due to the VNS removal versus the second procedure, and thus this data was unavailable.

### 2.2. Surgical Technique and Management

All operations were performed by the senior author (J.E.B.). The surgical technique involves first opening the chest wall incision, followed by the removal of the VNS generator. Next, adhesions are removed from the lead wire in order to ensure the free movement of the wire. Gentle pressure is applied to the anterior neck at the level of the helical coils while traction is applied inferiorly to the lead wire at the chest wall incision (Appendix A). Once the lead wire is removed, the neck is evaluated using intraoperative fluoroscopy, looking for residual wire (Figure 1). If the lead wire is completely removed, the chest incision is closed and the patient is extubated and transported to the recovery room. If fluoroscopy reveals that retained lead wire is present (Figure 1b), the neck incision is explored, and the residual lead wire removed. Lead wire removal is confirmed by intra-operative fluoroscopy. Both incisions are then closed, and the patient is extubated and transported to the recovery room.

Once the patient has cleared anesthesia, they are evaluated for vocal cord weakness. If present, they are kept NPO until a Speech Language Pathology-supervised swallow study is completed. If the swallow study is abnormal, dietary modifications, including a thickened diet, are initiated. Patients’ vocal cord weakness is followed until cleared by Speech Language Pathology.

### 2.3. Statistical Analysis

All statistical tests were performed in GraphPad Prism software (version 9.5.1). For binary data, means testing was performed using a two-tailed Fisher’s exact test based on a 2 × 2 contingency table. For continuous data, means testing was achieved via a two-tailed Student’s t-test, with assumptions of the homogeneity of variances and normality achieved via an F-test and Shapiro–Wilk test, respectively.

This study design has inherent limitations due to its retrospective nature, limited study population, lack of a prospective control, and the fact that the decision was made to attempt removal via single incision first in all cases. This strategy, however, is adequate for assessing basic questions pertaining to the safety, viability, and potential time reduction with this approach.

## 3. Results

### 3.1. Surgical Outcomes and Patient Characteristics

A total of 73 patients met the inclusion criteria. A single-incision removal was achieved in 48 cases (65.8%), whereas conversion to a two-incision approach was performed in 25 cases (32.9%). The VNS leads were successfully removed without residual lead wire in 71 cases (97.3%). Of these two remaining cases, one was a single-incision approach where residual wire was identified postoperatively, and the patient was returned to the OR and the remaining wire removed via a second incision. In the other case, the wire was left, due to an internal jugular vein injury that was unable to be repaired. There was no statistically significant difference (*p* = 0.8650, two-tailed Mann–Whitney U test) between the ages of the patients in the single-incision group (24.6 years, range: 3–55) and two-incision group (26.0, range: 8–61).

The time in the operating room for VNS removal was available for 35 patients (24 single-incision cases and 11 two-incision cases). Time in the operating room was 29.4 min (range 11–84) for the single-incision group, as opposed to 74.2 min (range 33–203) for the two-incision group.

### 3.2. Complications

Overall, 7 out of 48 (14.6%) patients in the single-incision group developed transient hoarseness, dysphagia, or cough, compared to 3 out of 25 (12.0%) patients in the two-incision group. This difference was not significant according to a two-tailed Fisher’s exact test (*p* = 0.7368). There were no cases of permanent deficits. For the single-incision group, there were no cases of internal jugular vein injury (0%), compared to two cases in the two-incision group (8.0%). Both injuries occurred as a consequence of the neck dissection. This difference did not achieve statistical significance (*p* = 0.1265, two-tailed Fisher’s exact test).

## 4. Discussion

While vagal nerve stimulation is often an effective palliative intervention for seizure management, VNS removal may be necessary due to lack of efficacy, interference with MRI-based procedures, and/or cosmetic concerns. This surgery is typically achieved via a two-incision approach, including a neck dissection for the removal of the lead wires. This approach carries a small but non-negligible risk of injury to vascular structures and to the recurrent laryngeal nerve, which may be permanent. This dissection is particularly challenging due to the often heavy scarring and loss of anatomic landmarks.

Here, we report our experience of a single-incision approach to VNS removal. The successful removal of the lead wires was achieved in the majority of cases (65.8%), thereby bypassing the need for neck dissection and possibly reducing the OR time. No cases of vascular injury or permanent recurrent laryngeal nerve injury occurred. The most common complications in this patient population were transient hoarseness, cough, or dysphagia, which likely reflect neurapraxia of the vagal nerve or recurrent laryngeal nerve. Notably, all patients underwent traction on the lead wires. The rates in this study are equivalent in the two groups. Thus, the equivalent rates of hoarseness, cough, or dysphagia may indicate that these side effects are primarily a result of this maneuver. The overall rate of dysfunction in this cohort (13.7%) is higher than that reported in the meta-analysis by Saba et al. [8]. (5.1%); however, there were no cases of permanent dysfunction in this cohort, as opposed to 1.6% in the meta-analysis.

A possible advantage of this approach is the reduced OR time, with the single-incision approach taking roughly half as long on average as the two-incision method. While there is an artificial component to this result, given that the single-incision approach was attempted first in all cases, both surgical approaches start equivalently, with the dissection of the generator. The attempted traction procedure to fracture the coils takes seconds and, if successful, allows one to skip the neck dissection. Thus, this result is highly likely to be consistent and reproducible. This approach further abrogates the need for a neck incision given the associated risk of neck dissection, wound healing issues, and surgical site infection. This approach may also carry a reduced risk of permanent laryngeal nerve injury and internal jugular injury, though more data are needed to validate this claim.

Furthermore, the viability of this approach may warrant changes in the design of the VNS wire, and it is worthwhile to consider the potential advantages of a wire that has a mechanism in place to be detached, which may reduce the rates of transient nerve injury in this population and increase the rates of successful single-incision removal.

## 5. Conclusions

VNS removal can be safely achieved via a single-incision method in the majority of cases. This approach results in a possibly reduced OR time, and no patients undergoing single-incision removal experienced permanent vocal cord dysfunction or vascular injury. This approach may increase the risk of transient hoarseness, dysphagia, and/or cough.

## Figures and Tables

**Figure 1 brainsci-15-00738-f001:**
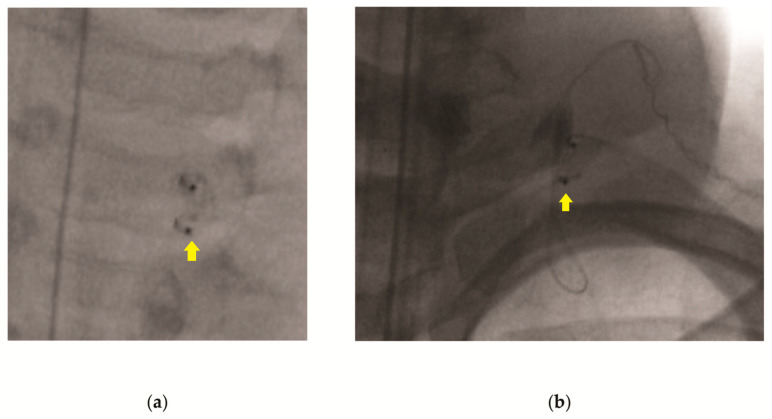
Radiographs showing coils and lead wire following attempted single-incision removal. (**a**) A-P radiograph showing residual coils but no wire; (**b**) lateral radiograph showing coils with residual guide wire. Yellow arrows indicate helical coils.

## Data Availability

The study data is available upon reasonable request.

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
