# Peer review of "A Single-Incision Method for the Removal of Vagus Nerve Stimulators: A Single-Institution Retrospective Review"

_brainsci, 2025, doi:10.3390/brainsci15070738_

Round 1
Reviewer 1 Report
Comments and Suggestions for Authors
Clarity that time of procedure was only available for 35 of 73. Of the 35, times were compared and conclusions made comparing 24 single incision against 11 double incision. All the double incisions started out to be single incision which makes this statistic misleading / irrelevant. Furthermore, the two-tailed Mann Whitney U test for statistical significance has to be taken with a grain of salt and possibly violates the null hypothesis. There is mention that there is an artificial component to this result (line 166), this does not address the small number involved.
The pictures (Figure 1) need arrows added to designate the coils which are difficult to see.
Finally, it should be a call to develop a semi-detachable lead at the coil which is intended to breakaway when pulled on at the level of the generator.
Author Response
Reviewer one:
Clarity that time of procedure was only available for 35 of 73. Of the 35, times were compared and conclusions made comparing 24 single incision against 11 double incision. All the double incisions started out to be single incision which makes this statistic misleading / irrelevant. Furthermore, the two-tailed Mann Whitney U test for statistical significance has to be taken with a grain of salt and possibly violates the null hypothesis. There is mention that there is an artificial component to this result (line 166), this does not address the small number involved.
We thank the reviewer for highlighting these issues, and we have deleted the test and clarified the limitations. We do, however, feel that this statistic is still worth including, given that a possible benefit of this procedure is reduced OR times, and a breakdown of how long the operation typically takes in a single-incision context is relevant to the reader.
Changes to text: Statistical test removed, text added: “though all cases initially began with the intention of attempting a single-incision ap-proach.” Lines 130-131.
The pictures (Figure 1) need arrows added to designate the coils which are difficult to see.
This has been added to the figure and figure legend.
Finally, it should be a call to develop a semi-detachable lead at the coil which is intended to breakaway when pulled on at the level of the generator.
We have added this to the discussion.
Changes to text: “The viability of this approach, furthermore, may warrant changes to the design of the VNS wire, and it is worthwhile to consider the potential advantages of a wire that has a mechanism in place to be detached, which may reduce the rates of transient nerve injury in this population.”
Reviewer 2 Report
Comments and Suggestions for Authors
The manuscript is methodologically sound, well organized, and follows a clear structure. The primary objectives and hypothesis are well-defined and based on measurable outcomes such as operating time and incidence of complications. I’d encourage the authors to explicitly state the inherent limitations due to a) lack of prospective control, b) limited study population, and c) the fact that the choice (or lack thereof) of starting with a single-incision method could potentially add bias.
Strengths:
• Clinically relevant topic in epilepsy management
• Good methodological rigor especially considering the retrospective nature of the study
• Appropriate use of statistical tests for group comparisons
Weaknesses:
• Limited statistical power owing to the insubstantial study population
• Need to address surgeon variability – same/multiple surgeons? performed the one/two-incision procedures?
Comments:
• Suggest revising the Introduction section to be more concise
• Line 60: Remove additional space character before period.
• Line 169: Misspelt “result” as “results”
• Consider reporting Median operating times as well
Author Response
Reviewer two:
The manuscript is methodologically sound, well organized, and follows a clear structure. The primary objectives and hypothesis are well-defined and based on measurable outcomes such as operating time and incidence of complications. I’d encourage the authors to explicitly state the inherent limitations due to a) lack of prospective control, b) limited study population, and c) the fact that the choice (or lack thereof) of starting with a single-incision method could potentially add bias.
We thank the reviewer for this comment. This information has been added to the methods section.
Changes to text: “This study design has inherent limitations owing to the retrospective nature, lim-ited study population, lack of a prospective control, and the fact that the decision was made to attempt removal by single incision first in all cases. This strategy, however, is adequate to assess basic questions pertaining to the safety, viability, and potential time reduction with this approach.” – line 109-113.
Strengths:
• Clinically relevant topic in epilepsy management
• Good methodological rigor especially considering the retrospective nature of the study
• Appropriate use of statistical tests for group comparisons
Weaknesses:
• Limited statistical power owing to the insubstantial study population
• Need to address surgeon variability – same/multiple surgeons? performed the one/two-incision procedures?
All operations were performed by the senior author, and this has been added to the text.
Changes to text: “All operations were performed by the senior author (JEB).” – line 87.
Reviewer 3 Report
Comments and Suggestions for Authors
This paper describs a new approach for removal of VN stimulators. A single chest incision to remove the generator and a traction of the wires at the level of the helical coil to disconnect the lead wire.
NB: Video1: it is impossible to display it.
Study design,surgical technique and results are clearly and syntetically presented as well as the complications. The discussion highlights both the significant reduction of the OR time and the advantages of the single incision. The neck incision being the more risky for neck dissection,wound healing issues and infection. The risk for laringeal nerve and for the internal jugular injuries are reduced
Author Response
Reviewer Three
This paper describs a new approach for removal of VN stimulators. A single chest incision to remove the generator and a traction of the wires at the level of the helical coil to disconnect the lead wire.
NB: Video1: it is impossible to display it.
We thank the reviewer. We have reuploaded the video.
Study design,surgical technique and results are clearly and syntetically presented as well as the complications. The discussion highlights both the significant reduction of the OR time and the advantages of the single incision. The neck incision being the more risky for neck dissection,wound healing issues and infection. The risk for laringeal nerve and for the internal jugular injuries are reduced
We thank the reviewer for their positive comments.